# Diversity of Bacterial Clones and Plasmids of NDM-1 Producing *Escherichia coli* Clinical Isolates in Central Greece

**DOI:** 10.3390/microorganisms11020516

**Published:** 2023-02-17

**Authors:** Katerina Tsilipounidaki, Zoi Florou, Anargyros Skoulakis, George C. Fthenakis, Vivi Miriagou, Efthymia Petinaki

**Affiliations:** 1Faculty of Medicine, University of Thessaly, 41500 Larissa, Greece; 2Veterinary Faculty, University of Thessaly, 43100 Karditsa, Greece; 3Laboratory of Bacteriology, Hellenic Pasteur Institute, 11521 Athens, Greece

**Keywords:** antibiotic resistance, *bla*
_NDM-1_, *Escherichia coli*, Greece

## Abstract

The objective of the present study was to genetically characterize ten NDM-1 producing *Escherichia coli* isolates, recovered from patients in a hospital in Central Greece during the period 2017 to 2021.The isolates were studied by whole genome sequencing to obtain multi-locus sequencing typing (MLST), identification of *bla*_NDM1_-environment, resistome and plasmid content. MLST analysis showed the presence of eight sequence types: ST46* (two isolates), ST46, ST744, ST998, ST410, ST224, ST4380, ST683 and ST12 (one isolate each). Apart of the presence of *bla*_NDM-1_, the isolates carried a combination of various to *β*-lactams encoding resistance genes: *bla*_TEM-1B_, bla_CTX-15_, *bla*_OXA-1_, *bla*_VIM-1_, *bla*_SHV-5_, *bla*_OXA-16_, *bla*_OXA-10_ and *bla*_VEB-1_. Additionally, plurality of resistance genes to aminoglycosides, macrolides, rifamycin, phenicols, sulfonamides and tetracycline was detected. The presence of multiple replicons was observed, with predominance of IncFII and IncFIB. Analysis of *bla*_NDM-1_ genetic environment of the isolates showed that seven had 100% identity with the pS-3002cz plasmid (Accession Number KJ 958927), two with the pB-3002cz plasmid (Accession Number KJ958926) and one with the pEc19397-131 plasmid (Accession Number MG878866). Τhis latter plasmid was derived by the fusion of two, previously identified, plasmids, pAMPD2 and pLK75 (Accession Numbers CP078058 and KJ440076, respectively). The diversity of clones and plasmids of NDM-1 producing *E. coli* isolated from patients in Greece indicates a continuous horizontal gene transfer.

## 1. Introduction

*Escherichia coli* is a bacterial species with high diversity, ranging from intestinal commensal strains to pathogenic strains causing urinary tract infection, acute enteritis, sepsis and neonatal meningitis [1]. In recent years, carbapenem-resistant *E. coli* (CREC) has been a serious problem worldwide, attracting significant clinical interest and attention [2,3]. As in carbapenem-resistant *Enterobacterales*, the acquisition of carbapenemase-encoding genes is also the most common mechanisms for CREC [3,4].

Three classes of carbapenemases have been identified: class A carbapenemase (*Klebsiella pneumoniae* carbapenemase-KPC), class B metallo-*β*-lactamases (Verona integron-encoded metallo-*β*-lactamase-VIM, New-Delhi metallo-*β*-lactamases-NDM, IMP *β*-lactamase-IMP) and class D enzymes (OXA-48 type) [5,6].

Recently, several reports have demonstrated the dissemination of *bla*_NDM_ encoding gene among *E. coli* isolated from patients in various geographical locations [7]. The heterogenicity of the genetic environment of *bla*_NDM_ encoding gene, combined with its location on plasmids of different incompatibility groups (Inc) has led to the emergence of diverse CREC clones worldwide [8,9].

In Greece, carbapenem-resistant *E. coli* isolates have been recovered rarely [10]. Two previous studies have reported the presence of *bla*_KPC_, *bla*_NDM-1_ and *bla*_NDM-5_ positive *E. coli* isolates, recovered from humans and animals [10,11]. Although clonality was observed among *bla*_KPC_ positive *E. coli*, primary data have revealed a diversity of clones among NDM-producing *E. coli* [10]. This finding has raised the questions, first if the same or different plasmids have been introduced into *E. coli* isolates, and second if the genetic context of *bl*a_NDM-1_ would be the same among the various clones.

The objective of the present study was to genetically characterize ten NDM-producing *E. coli* isolates, recovered from patients in Central Greece during the period 2017 to 2021.

## 2. Materials and Methods

### 2.1. Isolation of bla_NDM-1_ Escerichia coli

Ten (10) *bla*_NDM-1_ positive *E. coli* isolates were studied in this work. All isolates had been recovered from clinical samples of patients admitted to the University Hospital of Larissa in Central Greece, a tertiary-care hospital, during the period 2017 to 2021.

The identification and the susceptibility testing of the isolates were carried out using the automated Vitek-2 system (BioMerieux, Marcy l’ Etoile, France). Minimal Inhibitory Concentrations (MICs) of imipenem and meropenem were determined by MIC test strip (Liofilchem, Roseto degli Abruzzi, Italy); MIC to colistin was determined by broth microdilution method, following the respective EUCAST guidelines (https://www.eucast.org, accessed on 10 October 2022). All the isolates were tested for phenotypic production of carbapenemase, by using MIC strips containing meropenem plus ethylenediaminetetraacetic acid (EDTA) and meropenem plus phenylboronic acid (Liofilchem).

Isolates found with a ratio meropenem/(meropenem plus EDTA) ≥ 8 were subsequently tested for detection of the carbapenemase encoding genes *bla*_VIM_, *bla*_NDM_, by means of a relevant PCR followed by sequencing analysis [10]. The ten (10) *bla*_NDM-1_ positive *E. coli* were identified and were further studied by Whole Genome Sequencing (WGS) analysis.

### 2.2. Whole Genome Sequencing of bla_NDM-1_ Positive E. coli

Initially, libraries were prepared using Ion Torrent technology and Ion Chef work flows (Thermo Fisher Scientific, Waltham, MA, USA). Genomic DNA libraries were sequenced on the S5XLS system and analysis of primary data was conducted with Ion Torrent Suite v.5.10.0 (Thermo Fisher Scientific). The quality of the reads was checked using FastQC software version 0.11.9. The reads for each sample were assembled using the SPAdes genome assembler v3.15.5 with the default parameters. The quality of the assembled genomes was assessed with the tool Quast version 5.2.0. The average coverage for each genome was computed using the tool mapPacBio from BBTools (https://sourceforge.net/projects/bbmap/, accessed on 25 October 2022). Gaps were filled by sequencing of overlapping PCR produced fragments. 

Typing of isolates, based on the Achtman scheme, was assessed by using the online tool MLST 2.0. Evaluation for the presence of antibiotic resistance genes in the assembled genomes was performed by using the online tool ResFinder-4.1, with the ID threshold set to 90% and the minimum length set to 60%. The presence of plasmids in the assemblies was assessed by means of Plasmid Finder v2.1, using the *Enterobacterales* database, with the minimum identity threshold set to 95% and the minimum coverage set to 60%. In order to determine the origin of the genetic contexts of *bla*_NDM-1_ encoding gene, a BLAST analysis was performed. Only results with a high identity score (100% identity and ≥90% cover age) were considered.

### 2.3. Nucleotide Sequence Accession Numbers

The genomes of the *bla_N_*_DM-1_ positive *E. coli* strains 9703, 9793, A155, A1160, A1630, A2614, A3025, A3026, A3039 and A3040, included in the present study, were deposited in GenBank under accession numbers: GCA_026421085.1, GCA_026421065.1, GCA_026421025.1, GCA_026421005.1, GCA_026421045.1, GCA_026420985.1, GCA_026420965.1, GCA_026420925.1, GCA_026420945.1, GCA_026420885.1, respectively.

### 2.4. Plasmid Analysis of bla_NDM-1_ Klebsiella Pneumoniae Strains

The *bla*_NDM-1_ encoding plasmids identified in *E. coli* isolates were compared to plasmids found among *K. pneumoniae* population. Five *bla*_NDM-1_ positive *K. pneumoniae* strains, all of them recovered during the study period (2017–2021) at the University Hospital of Larissa, were also selected for plasmid analysis. Selection was based on their ST types and the presence of carbapenemase-encoding genes. Two of these strains (B3118 and B3119), belonging to ST11, carried both *bla*_NDM-1_ and *bla*_OXA-48_ encoding genes, one strain (B3185), belonging to ST11, carried both *bla*_NDM-1_ and *bla*_KPC_ encoding genes, whilst the other two strains, belonging to ST11 (B3173) and ST231 (B3214), carried only the *bla*_NDM-1_ encoding gene. Whole genome analysis of those strains had been performed as described above (Section 2.2). 

### 2.5. Data Management

Data were entered into Microsoft Excel and analysed using SPSS v. 26 (IBM Analytics, Armonk, NY, USA). Initially, descriptive analysis was performed. Then, comparisons were performed by employing Pearson’s chi-square or Fisher exact test, as appropriate, and analysis of variance, according to the type of data. Statistical significance was defined at *p* < 0.05.

## 3. Results

### 3.1. Antimicrobial Susceptibility Profiles

All ten (100.0%) *bla_NDM-1_* positive *E. coli* were found to be resistant to imipenem (MIC > 16 mg L^−1^). Eight isolates (80.0%) were found to be additionally resistant to meropenem (>16 mg L^−1^); the remaining two isolates had MICs to meropenem 4 mg L^−1^ and thus were characterized as susceptible-increased exposure. Resistance rates to cefotaxime, ceftazidime, cefepime, aztreonam, gentamicin, amikacin, ciprofloxacin, tigecycline and colistin were 100.0%, 100.0%, 90.0%, 60.0%, 30.0%, 40.0%, 80.0%, 0.0% and 0.0%, respectively.

### 3.2. Multi-Locus Sequence Typing (MLST)

In silico performed MLST revealed that the ten isolates belonged to nine distinct Sequence Types (STs), specifically one isolate in each of ST744, ST998, ST410, ST224, ST4380, ST46, ST683, ST12, whilst two isolates were found to belong to a new type. This included a one base difference from *adk* 840 allele (^535^C → T), while the alleles of the other six genes were identical to those of ST46; it is noted as ST46*.

### 3.3. Identification of Resistance Genes

WGS analysis revealed that the ten *bla*_NDM-1_ positive *E. coli* strains cumulatively possessed 40 distinct resistance genes. Cumulatively, the ten strains possessed nine distinct genes encoding resistance to *β*-lactams and 13 distinct genes encoding resistance to aminoglycosides (i.e., 58.5% of distinct genes found, encoded resistance against one of these two antibiotic classes).

Median number of genes per strain was 14 (min.: 8 – max.: 20). Apart from *bla*_NDM-1_, no other β-lactamase encoding gene was detected in all ten strains. Other frequently detected genes were *aph(3″)-Ib*, *aph(6)-Id* and *sul2* (in eight strains each), as well as *bla*_TEM-1B_ and *tetA* (in seven strains each). Nine strains carried at least three genes encoding resistance to *β*-lactams (median number of such genes per strain = 3) and eight of these at least three genes encoding resistance to aminoglycosides (median number of such genes per strain = 4). These higher per strain numbers of genes encoding resistance against *β*-lactams or aminoglycosides were significant (*p* < 0.0001). 

All strains carried genes conferring resistance to at least six antimicrobial agents, whilst three strains carried genes against eight antimicrobial agents. A summary of the genes identified in the ten strains is in Figure 1.

### 3.4. Identification of Plasmids

Analysis of replicon sequences revealed that the ten strains carried 20 types of plasmids. Among these, IncFIB predominated (in eight strains), followed by IncI1-I(Alpha) and IncC (in seven strains each); IncFII was also frequently detected in five strains.

Median number of plasmids per strain was 6 (min.: 3–max.: 9). In nine strains, at least four and in seven strains at least six replicons were identified. 

Finally, three distinct Col plasmids were also detected. A summary of the replicons identified in the ten strains is in Figure 1.

### 3.5. Genetic Environment of bla_NDM-1_

A sequence comparison analysis revealed that three different *bla*_NDM-1_ genetic environments were identified among the strains studied (Figure 2). Seven strains (9793, A155, A1160, A3025, A3026, A3039 and A3040) shared a *bla*_NDM-1_ genetic context identical to that of plasmid pS-3002cz (Accession Number KJ 958927) [12]. Two other strains (9703 and A1630) had genetic context identical to that of plasmid pB-3002cz (Accession Number KJ958926) [12]. Plasmid analyses showed that *bla*_NDM-1_ encoding plasmids were identical to pS-3002cz and pB-3002cz, respectively [12].

The remaining tenth *E. coli* (2614) had a *bla*_NDM-1_ genetic context and plasmid, both identical with plasmid pEc19397-131 (Accession Number MG878866). In silico analysis of pEc19397-131 showed that this had derived by fusion of two, previously identified, plasmids, specifically pAMPD2 and pLK75 (Accession Numbers CP078058 and KJ440076, respectively) [13].

Finally, plasmid analysis of *K. pneumonia*e revealed that the same plasmids were circulating among *K. pneumoniae* isolates during that period. Four of the five strains carried plasmid pS-3002cz (B3118, B3119, B3173, B3185) and one strain (B3214) also carried plasmid pB-3002cz with 100% identity Accession Numbers GCA_028593925.1, GCA_028593815.1, GCA_028593845.1, GCA_028595105.1, GCA_028593905.1, respectively.

## 4. Discussion

The first description of *bla*_NDM-1_ was published in 2009 [14]. *K. pneumoniae* and *E. coli* are the most commonly described NDM-1 producing bacteria. However, the *bla*_NDM-1_ encoding gene has also been detected in *Enterobacterales* other than *K. pneumoniae* and *E. coli* [15]; NDM-1 production has been also found in clinical isolates of *Acinetobacter baumannii* and *Pseudomonas aeruginosa*, as well as in a wide variety of other, non-fermenting Gram-negative bacteria [16].

Until 2017, no *bla*_NDM-1_ positive *E. coli* isolates had been reported from Greece; in contrast, several studies had indicated the dissemination of *bla*_NDM-1_ positive *K. pneumoniae* in many hospitals of the country; in a large multicenter study, conducted between 2013 to 2016 across the country, 71% of *K. pneumoniae* isolates found to be phenotypically MBL positive, were also found to be *bla*_NDM-1_ positive, with ST11 being the predominant clone [17,18,19,20]. The same epidemiological findings, with little differences regarding *bla_NDM-1_* positive *K. pneumoniae* frequency and clonality, are observed up today in the country.

The first three *bla***_NDM-1_** positive *E. coli* strains in the country were recognized in 2017 at the University Hospital of Larissa, a tertiary care hospital located in Central Greece [10]. During the following years (2018–2021), another seven isolates have been recognized from clinical specimens collected from patients in the hospital. The proportion of *bla*_NDM-1_ positive *E. coli* has remained stable throughout the study period, corresponding to 0.17% of total *E. coli* isolates recovered annually.

The dissemination of *bla*_NDM-1_ encoding gene among *K. pneumoniae* and *E. coli* is carried out via plasmids or other mobile genetic elements. Previous studies have shown the presence of two plasmids, pB-3002cz and pS-3002cz, both previously identified in the Czech Republic, among the *bla*_NDM-1_ positive *K. pneumoniae* strains isolated in the country [17,18,19]; however, no data have been reported about the characterization of plasmids among *bla*_NDM-1_
*E. coli.* The above two plasmids were carried by nine of the *bla*_NDM-1_ positive *E. coli* studied in the present work. Given that the *bla*_NDM-1_ positive *K. pneumoniae* previously recovered, also carried the same plasmids, the present findings indicate *bla*_NDM_ encoding gene’ transmission between different bacterial species and different clones of the same species.

Only one *bla*_NDM-1_ positive *E. coli* studied in the present work, was found to carry an uncommon plasmid, previously found in a strain recovered in China in 2018. Clinical information from the Greek patient from whom this strain had been isolated, revealed no travel abroad and no contact with people with a travel history.

A literature search at the NCBI database (last consulted on 28 December 2022) revealed that over 750 different plasmids might carry the *bla*_NDM-1_ encoding gene in *E. coli* [21,22,23]. Recently, a genomic epidemiological study of global carbapenemase-producing *E. coli* conducted between 2015–2017 in 62 countries showed that NDM-1 was the most frequent NDM-enzyme and was associated with various STs within diverse plasmids [24]; 13 different STs were identified obtained from Egypt, Morocco, Serbia, Romania, Guatemala, Kuwait, Philippines, Russia, Thailand and Vietnam [25]. In addition, in Lebanon, NDM-1 producing *E. coli* have been described since 2012 [26,27,28]. These data indicate that *bla*_NDM-1_ positive *E. coli* were circulating in neighbouring countries such as Serbia, Romania, Lebanon, Morocco and Egypt prior of their identification in Greece.

NDM lactamases cause high-level resistance to many clinically available *β*-lactams and are not inhibited by the novel *β*-lactamase inhibitors, e.g., avibactam, relebactam or vaborbactam. Although the novel catecholamine-siderophore cephalosporin cefiderocol can be an option for treatment of infections caused by NDM-producing *E. coli,* a recent study has revealed already emergence of cefiderocol-resistant NDM-5 producing *E. coli* [29]. These findings further support the need for continuous surveillance of NDM-producing *E. coli* strains. 

## Figures and Tables

**Figure 1 microorganisms-11-00516-f001:**
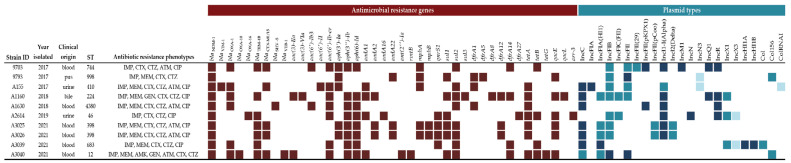
Visual presentation of antimicrobial resistance genes and Inc plasmid types detected in ten *bla*_NDM-1_ positive *Escherichia coli* strains, as found during whole genome sequence In the plasmid type section different colors indicate the percentage of genetic similarity according to BlastN matching with previously published plasmids in the NCBI database: Dark blue ≥ 99%, Blue: 90–98%, Light Blue: 80–89%. IMP: Imipenem, MEM: Meropenem, CTX: Cefotaxime, CTZ: Ceftazidime, FEP: Cefepime, ATM: Aztreonam, GEN: Gentamicin, AMK: Amikacin, CIP: Ciprofloxacin.

**Figure 2 microorganisms-11-00516-f002:**
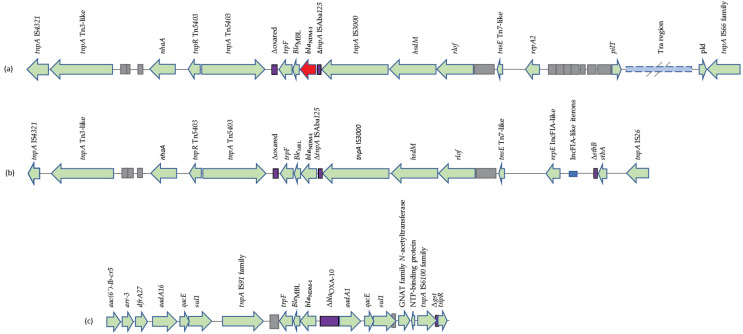
Presentation of different *bla*_NDM-1_ genetic context. (**a**) 9703 and A1630 identical to plasmid pB-3002cz (Accession Number KJ958926); (**b**) 9793, A155, A1160, A3025, A3026, A3039 and A3040 identical to plasmid pS-3002cz (Accession Number KJ958927); (**c**) A2614 identical to plasmid pEc19397-131 (Accession Number MG878866).

## Data Availability

Not applicable.

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
