# Peer review of "Diversity of Bacterial Clones and Plasmids of NDM-1 Producing *Escherichia coli* Clinical Isolates in Central Greece"

_microorganisms, 2023, doi:10.3390/microorganisms11020516_

Round 1

Reviewer 1 Report

The authors provided us with a quite interesting results but should pay more attention to quality of writing/text editing, e.g.: lines 11, 12, 75, 89, 91, 178.

Moreover:

Line 12 - was it gene or genome sequencing?

Line 59 and 60 "MIC to antimicrobials" sound awkward.

Line 115 - "fully resistant" - what does it mean? One should never use this phrase.

Line 126 - the authors should define a certain criteria for genes/genomes differentiation - should it be underlined/confirmed that the noted mutation is a SNP rather, not a new gene type/STs?

Table 1 and Figure 1 present basically a bunch of the same results, one should avoid it.

Line 163 - "Col plasmids"?

Line 165, 173 and the following - what are/is the mentioned "contexts/context"?

Line 179 - "klebsiellas" does not look like a scientific language.

Nevertheless, the authors presented the results that are quite interesting, especially on a local scale.

Author Response

Reviewer 1:

Thank you very much for your comments.

Line 12: the gene sequencing was replaced by genome sequencing

Line 59 and 60: MIC to was replaced by MIC of

Line 115: fully resistant was corrected as resistant

Line 126: as mentioned in the text one base difference from adk 840 allele (535C → T) was observed in two strains. The MLST scheme used was of Achtman; according to the scheme of Pasteur the isolated belonged to ST398.

Table 1 was deleted.

Line 163: The nomenclature ‘’Col plasmid’’ exist. Please inform us if something else is needed.

Line 165, 173 and the following: “blaNDM-1 genetic context(s)” are the genetic platforms/gene arrays carrying the blaNDM-1 gene as described in the paper Sally Partridge & Jonathan R. Iredell, 2012, AAC 56(11), 6065-6067.  However, it has been replaced by “genetic environment”.

Line 179: klebsiellas was replaced by K. pneumoniae

Reviewer 2 Report

Specific remarks:

line 12 – not clear if it is a “whole gene” or a “whole genome”. Usually, MLST is performed by using data from whole genome sequencing.

line 19 – a plasmid is always a replicon, so “replicon plasmids” is a non-sense as there are no “no-replicon plasmids.”

line 20-21 – a short sentence about the similarity should be added

line 23  – every species is continuously evolving, while this study shows horizontal gene transfer and not evolution

line 40 - a plasmid is always a replicon, so “replicon plasmids” is a non-sense as there are no “no-replicon plasmids.” The “Inc” stands from the “incompatibility group”. The sentence should be rewritten.

line 106 – Genbank accession numbers should be cited.

lines 125-127 – a contradiction with the statement within the abstract

line 157-158, 161 – again wrong usge of the molecular genetics terminology.

line 214 – There is no such term as “horizontal mobile elements”. It should be replaced by “mobile genetic elements”, which can cause horizontal gene transfer.

General remarks:

Although English is not my mother tongue, I think a little language editing is needed, especially concerning punctuation.

I would like also to ask if the PlasmidSPAdes assembler was tried in order to reconstitute the plasmid sequences.

A good idea will be to compare the incidence and the first reports of NDM-1-positive E. coli in the neighboring countries

Author Response

Reviewer 2:

Thank you very much for your comments,

Line 12: the “gene sequencing” was replaced by “genome sequencing”

Line 19: it has been replaced by “replicons”

Line 20-21: a short sentence about the similarity shoud be added. It was added

Line 40: it has been replaced by “replicons .“Incompatibility group” has been added. The sentence was corrected/

Line 106: We have received the followed   Accession

We are currently processing your WGS submission for:

SUBID             BioProject        BioSample        Accession        Organism
---------------------------------------------------
SUB12698523        PRJNA893175        SAMN33014194        JAQQWW000000000        Klebsiella pneumoniae B3214
SUB12698523        PRJNA893175        SAMN33014193        JAQQWX000000000        Klebsiella pneumoniae B3185
SUB12698523        PRJNA893175        SAMN33014192        JAQQWY000000000        Klebsiella pneumoniae B3173
SUB12698523        PRJNA893175        SAMN33014191        JAQQWZ000000000        Klebsiella pneumoniae B3119
SUB12698523        PRJNA893175        SAMN33014190        JAQQXA000000000        Klebsiella pneumoniae B3118ssion

Now we are waiting for the final Accession Numbers.

Line 125-127: it was corrected

Line 157-158, 161: it was corrected

Line 214: It was corrected

General remarks:

I would like also ask if the PlasmidSPAdes assembler was tried in order to reconstttute the plasmid sequences.

The PlasmidSPAdes assembler was not used since we filled the gaps between the contigs/scaffolds by PCR and sequencing.

A good idea will be to compare the incidence and the first reports of NDM-1-positive E.coli in the neighboring countries.

A paragraph has been added (lines228-236)

Round 2

Reviewer 2 Report

I recognize that the authors complied with my suggestions so, I would recommend the publication of the manuscript in its current form.